# Granulomatosis with Polyangiitis Restricted to the Back Muscle: The First Case Report

**DOI:** 10.3390/medicina56090431

**Published:** 2020-08-27

**Authors:** Yu-Mi Lee, Sang Wan Chung

**Affiliations:** 1Division of Infectious Diseases, Department of Internal Medicine, Kyung Hee University Hospital, Kyung Hee University School of Medicine, Kyung Hee University, 26 Kyungheedae-ro, Dongdaemun-gu, Seoul 02447, Korea; cristal156@hanmail.net; 2Division of Rheumatology, Department of Internal Medicine, School of Medicine, Kyung Hee University, 26 Kyungheedae-ro, Dongdaemun-gu, Seoul 02447, Korea

**Keywords:** ANCA-associated vasculitis, organ limited vasculitis, GPA

## Abstract

Granulomatosis with polyangiitis (GPA) is an autoimmune disease which is a type of antineutrophil cytoplasmic antibody (ANCA)-associated vasculitis that frequently affects the lungs and kidneys. However, GPA limited to a single organ has also been reported. A 71-year-old man was admitted for back pain and fever. We detected elevated levels of inflammatory markers and myeloperoxidase-ANCA. Magnetic resonance imaging indicated diffuse inflammation of the back and psoas muscles. Histology showed degenerated muscle fibers and granulomatosis vasculitis with mixed lymphoplasma cell infiltration. High-dose methylprednisolone therapy improved his symptoms. A final diagnosis of GPA limited to the muscles was made.

## 1. Introduction

Granulomatosis with polyangiitis (GPA, previously known as Wegener’s granulomatosis) is an autoimmune disease which is a type of antineutrophil cytoplasmic antibody (ANCA)-associated vasculitis that affects small- to medium-sized vessels [1]. GPA can present with various signs and symptoms. GPA commonly affects the upper and lower respiratory tracts and kidneys, and is associated with otorhinolaryngological and renal manifestations [2]. However, numerous untypical manifestations may also occur. Such heterogeneity in clinical manifestations may be misleading and, as a consequence, delay establishment of proper diagnosis and treatment. Herein, we report the first case of ANCA-associated vasculitis restricted to the back muscle.

## 2. Presentation of Case Report

A 71-year-old male patient presented to the emergency room of Kyung Hee Medical Center due to fever and cramping pain in both legs lasting 4 weeks. The patient had been taking oral medications for diabetes mellitus (DM). He had chronic back pain due to spinal stenosis, and had received percutaneous epidural neuroplasty 8 days ago. He denied arthralgia on other peripheral joints and peripheral numbness. He had a fever of up to 38.5 °C without any skin rashes. He had no respiratory symptoms such as cough, sputum, or hemoptysis. A physical examination showed no remarkable findings except for tenderness at his lower back. His motor power was normal. His initial chest X-ray was normal. In terms of laboratory examinations, white blood cell count was 17.04 × 10^9^/L (76% neutrophil), erythrocyte sedimentation rate was 112 mm/h, and C-reactive protein (CRP) level was 137.6 nmol/L. Serum creatinine level (0.70 mg/dL), estimated glomerular filtration rate (GFR, 118.42 mL/min per 1.73 m^2^), and blood urine nitrogen level (19 mg/dL) were within normal range. His creatinine kinase (CK) level was normal, and no electrolyte disorder was identified. His aldolase level was 13.0 U/L (normal range, <7.6 U/L). Routine urinalysis revealed normal results. Due to his back pain and a history of percutaneous epidural neuroplasty, magnetic resonance imaging (MRI) of his lumbar spine was taken for the evaluation of infectious spondylitis. MRI showed diffuse inflammation of the back muscle at the lumbar spine level and of both psoas muscles at the L3-S1 level without abscess formation (Figure 1). Electrophysiological tests revealed a right lumbosacral polyradiculopathy at the L2-S1 level with no evidence of inflammatory myopathy. The myeloperoxidase (MPO)-ANCA test result was positive (3.9 index). A biopsy of the back muscle was performed. The pathologic results demonstrated mixed lymphoplasma cells, neutrophils, and eosinophils infiltration of the perivascular and vascular tissues with granulomas, which suggested granulomatous vasculitis (Figure 2A). CD68-positive cells were present in the immunohistochemistry of the back muscle (Figure 2B). By considering the results of MPO-ANCA and the pathological profile of his back muscle, he was diagnosed with GPA-associated myopathy. The patient received steroid pulse therapy (methylprednisolone, 1 g per day for 3 days) on day 30 of hospitalization, followed by methotrexate (12.5 mg per week) with steroid maintenance therapy. The patient became apyretic after steroid therapy, and CRP level decreased to 17.8 nmol/L. His back pain and bilateral leg cramping subsided slowly. The patient had an uneventful course without any complications for six months.

This study was approved by the Institutional Review Board (IRB) of Kyung Hee Medical Center, and the requirement for informed consent was waived (IRB 2020-07-037).

## 3. Discussion

We report a case involving an atypical presentation of vasculitis that affected the back muscle without evidence of other major organ involvement. High-dose steroid therapy was successful in controlling the disease. Infectious myopathy or infectious spondylitis was initially suspected because of his fever and back pain. The patient had a history of DM, and DM is one of the risk factors of infectious spondylitis [3]. However, despite receiving adequate antibiotic treatment, his symptoms did not improve. MRI is the standard imaging modality in the diagnosis of spinal disease; in this patient, there was no evidence of pyogenic infectious spondylitis or infectious myositis. MRI revealed diffuse inflammation in the back muscles at the L-spine level and in both psoas muscles at the L3-S1 level. There was no evidence of fluid collection. As an alternative diagnosis, we considered systemic rheumatic disease, and through ANCA titers, increased MPO-ANCAs were shown. Muscle biopsy was done for the evaluation of inflammatory myositis, and a definite diagnosis of GPA-associated myopathy was achieved. Complete remission was accomplished by steroids.

ANCA-associated vasculitis is a group of systemic vasculitis that are characterized by inflammation of the small vessels, ranging from capillaries to arterioles or venules. ANCA-associated vasculitis can be divided into three variants based on histological findings and clinical manifestations: microscopic polyangiitis (MPA), GPA, and eosinophilic GPA [4,5]. GPA preferentially affects the respiratory tract from the bronchus to the nasal cavity. GPA can also involve the kidneys, but to a lesser extent than in MPA. ANCA appears to be closely related to ANCA-associated vasculitis, but not all ANCA-associated vasculitis patients have ANCA, and the detection rate of ANCA has been lower than expected: MPO-ANCA in 30~80% of MPA patients and 0~10% of GPA patients [6,7]. While various clinical classification criteria have been developed, proposed, and validated, diagnostic differentiation between MPA and GPA is not as well-established. A clinical picture involving a positive ANCA test result and a positive biopsy result with evidence of necrotizing vasculitis, necrotizing glomerulonephritis, or granulomatous inflammation strongly suggests a diagnosis of GPA [8].

GPA can present with various signs and symptoms. Articular or muscular abnormalities have been reported at prevalence ranging from 4.7% to 67% [2,9], with the main complaint being arthralgia of the knees, hips, wrists, or ankles, while muscular involvement includes myalgia, weakness, and muscle enzyme elevation [10]. Several studies have reported vasculitis limited to the muscle. Khellaf et al. reported 11 cases of lower limb-restricted vasculitis [11]. Benz et al. also reported three cases of muscle-limited vasculitis [12]. However, back muscle involvement has not been documented. Previous reports have shown the success of immunosuppressive treatment (steroid and/or cyclophosphamide) in controlling the disease. Since vasculitis limited to the muscle is extremely rare, we were unable to anticipate the prognosis of this case. Careful observation for relapse of the disease is needed.

## 4. Conclusions

This case is helpful towards understanding the untypical manifestations of GPA to facilitate early diagnosis and proper management. Primary vasculitis of the muscle should be considered in patients presenting with pain and signs of systemic inflammation in the absence of other organ involvement.

## Figures and Tables

**Figure 1 medicina-56-00431-f001:**
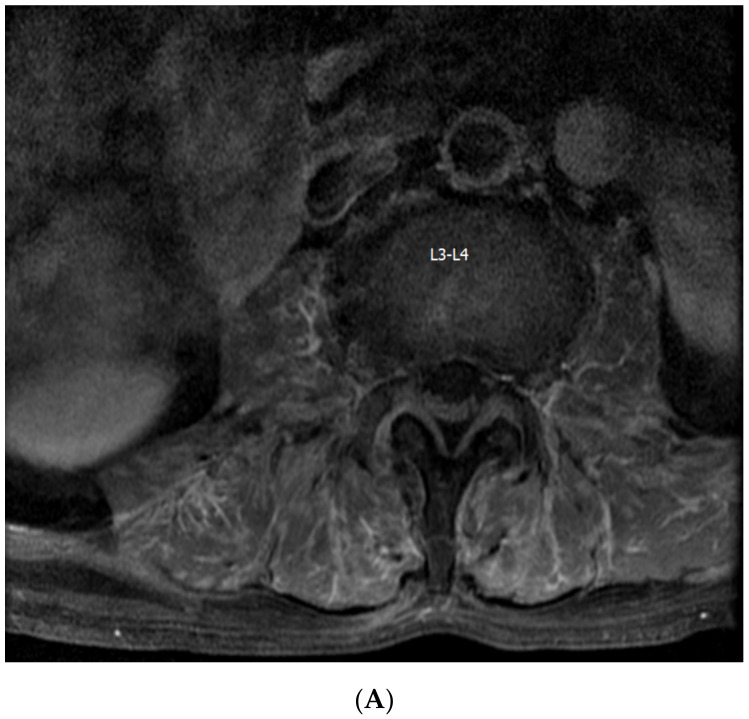
Lumbar spine magnetic resonance imaging (MRI). (**A**) Fast spin-echo T1-weighted MRI and (**B**) T2-weighted MRI. Lumbar spine MRI showed diffuse inflammation in the back muscle at the lumbar 3-sacral 1 level.

**Figure 2 medicina-56-00431-f002:**
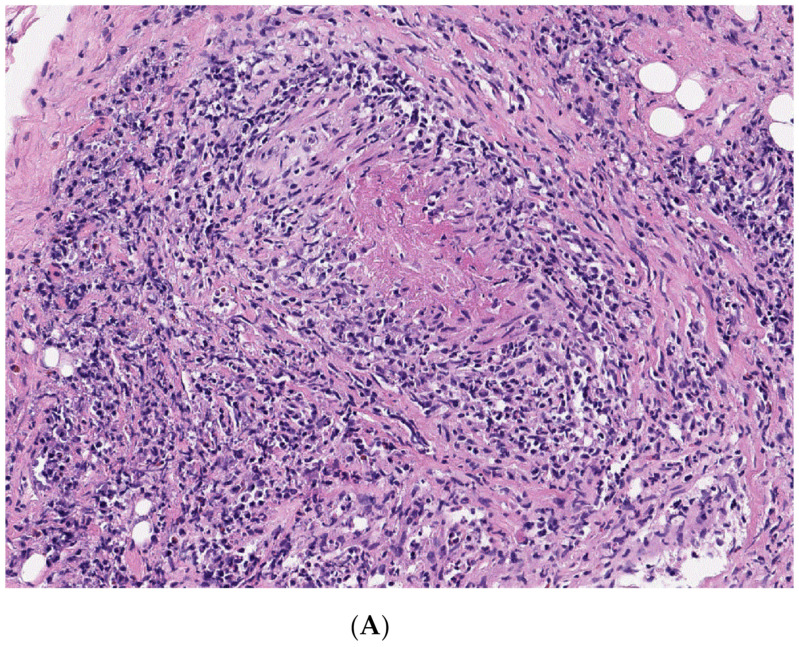
Pathologic findings of the back muscle at the lumbar level L3–4. (**A**) Hematoxylin and eosin stain, 200×. The back muscle biopsy indicated infiltration of mixed lymphoplasma cells, and neutrophils in perivascular and vascular tissues with granulomas, concordant with granulomatosis with polyangiitis (GPA). (**B**) Immunohistochemical stain for CD68 pgm1, 200×. Many CD68 positive cells were present.

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
