# Peer review of "Granulomatosis with Polyangiitis Restricted to the Back Muscle: The First Case Report"

_medicina, 2020, doi:10.3390/medicina56090431_

Round 1
Reviewer 1 Report
Great and interesting case, well written.
few comments, to highlight absence of other organ involvement by history and physical exam. Any evidence of bone or cartilage destruction , any sinusitis, absence of cough/pulmonary symptoms, nasal bleeding, hemoptysis. UA and absence of hematuria, absence of neurological symptoms.
the reason I mentioned Minor is because in the case report it's mentioned that back muscle involvement was the only finding and they didn't write specifically what other organ systems were addressed such as absence of sinusitis/lung/kidney involvement, provide absence of nasal crusting, present labs including kidney function. instead of just saying ( no other organ involvement)
Author Response
Thanks to carefully review the manuscript of the author.
As you recommended, we added sentences about the absence of other organ involvement as below
Page 3, Line55-57
He denied arthralgia on other peripheral joints and peripheral numbness. He had a fever of up to 38.5℃ without any skin rashes. He had no respiratory symptoms such as cough, sputum or hemoptysis.
Page 3, Line 65
Routine urinalysis revealed normal results.
Reviewer 2 Report
Thank you for your case report.
It would be interesting to see whether a follow-up MRI was done, if so, if it revealed improvement, and whether such improvement coincided with clinical improvement or lagged behind clinical improvement.
Author Response
Thanks to carefully review the manuscript of the author.
Unfortunately, no follow-up MRI was done. There are many reasons, one of them is a cost issue, the response to treatment in inflammatory myositis is judged by clinical features or laboratory test.